# 14-3-3 Activated Bacterial Exotoxins AexT and ExoT Share Actin and the SH2 Domains of CRK Proteins as Targets for ADP-Ribosylation

**DOI:** 10.3390/pathogens11121497

**Published:** 2022-12-08

**Authors:** Carmen Ebenwaldner, Peter Hornyak, Antonio Ginés García-Saura, Archimede Torretta, Saber Anoosheh, Anders Hofer, Herwig Schüler

**Affiliations:** 1Center for Molecular Protein Science (CMPS), Department of Chemistry, Lund University, 22100 Lund, Sweden; 2Department of Biosciences, Karolinska Institutet, 14157 Huddinge, Sweden; 3Department of Medical Biochemistry and Biophysics, Umeå University, 90736 Umeå, Sweden

**Keywords:** 14-3-3 activated bacterial exotoxins, actin, ADP-ribosylation, *Aeromonas salmonicid*, protein refolding, type III secretion system, *Pseudomonas aeruginosa*

## Abstract

Bacterial exotoxins with ADP-ribosyltransferase activity can be divided into distinct clades based on their domain organization. Exotoxins from several clades are known to modify actin at Arg177; but of the 14-3-3 dependent exotoxins only *Aeromonas salmonicida* exoenzyme T (AexT) has been reported to ADP-ribosylate actin. Given the extensive similarity among the 14-3-3 dependent exotoxins, we initiated a structural and biochemical comparison of these proteins. Structural modeling of AexT indicated a target binding site that shared homology with *Pseudomonas aeruginosa* Exoenzyme T (ExoT) but not with Exoenzyme S (ExoS). Biochemical analyses confirmed that the catalytic activities of both exotoxins were stimulated by agmatine, indicating that they ADP-ribosylate arginine residues in their targets. Side-by-side comparison of target protein modification showed that AexT had activity toward the SH2 domain of the Crk-like protein (CRKL), a known target for ExoT. We found that both AexT and ExoT ADP-ribosylated actin and in both cases, the modification compromised actin polymerization. Our results indicate that AexT and ExoT are functional homologs that affect cytoskeletal integrity via actin and signaling pathways to the cytoskeleton.

## 1. Introduction

*Aeromonas* and *Pseudomonas* are ubiquitous Gram-negative bacteria that can cause infections in a range of tissues. The planktonic *Aeromonas* species are opportunistic pathogens of diverse aquatic animals; they can display resistance to multiple antibiotics and can cause serious economic damage to fisheries [1]. The most common clinical manifestation of aeromonads is seafoodborne gastroenteritis caused by *Aeromonas salmonicida*. Both aeromonads and pseudomonads cause difficult to treat wound and airway infections [2,3,4], with pseudomonads being among the principal causative agents of nosocomial infections [5].

Contact with eukaryotic target cells triggers signalling cascades in *Pseudomonas aeruginosa* leading to the expression of the transcription factor ExsA, which in turn regulates the expression of all type III secretion system (T3SS) genes [6]. These genes encode the structural components of the T3SS needle apparatus as well as four exotoxins [7], and their expression is the major determinant of virulence in *Pseudomonas* [8]. Expression of the ADP-ribosyltransferase (ART) toxin exotoxin-S (exoenzyme-S; ExoS) and the phospholipase ExoU is mutually exclusive in nearly all *Pseudomonas* isolates examined [8,9]. A second ART exotoxin, ExoT, is expressed in a majority of clinical isolates [9]. Presence of either ExoS or ExoT has been shown to be sufficient to induce morphological changes in infected cultured cell lines [10,11]. In addition, ExoT stimulates apoptosis in infected host cells [12]. In *A. salmonicida* AexT, an ortholog of ExoS and ExoT, is also injected via a T3SS; AexT appears to be the major determinant of virulence in *Aeromonas* strains [13]. Fish gonad RTG-2 cells infected with *A. salmonicida* undergo cell rounding whereas infection with a mutant strain devoid of AexT does not affect cell morphology [13].

Generally, ADP-ribosyl transferases catalyze the formation of a posttranslational modification that affects or determines the activities and the fates of a multitude of proteins within human cells [14]. Diverse pathogenic bacteria have evolved mechanisms to employ ADP-ribosylation in generating a replication competent niche for themselves. ADP-ribosylating toxins that target eEF-2, trimeric G-proteins, as well as actin and other cytoskeletal proteins and their regulators including small GTPases have been identified [15,16,17]. A distinct clade of ADP-ribosylating bacterial toxins, with *P. aeruginosa* ExoS as their founding member, get activated by binding to host-derived 14-3-3 proteins upon their T3SS-dependent delivery into the host cytosol [18]. Most members of this clade, including *Pseudomonas* ExoT [19] and *Aeromonas* AexT [20], combine a GTPase activating protein (GAP) domain with an ART domain [21]. Whereas the GAP domains of ExoS, ExoT and AexT appear to target the same host GTPases, the ART domain specificities differ between the three exotoxins [7,19,20,22,23,24]. ExoS has been shown to affect host cell morphology by ADP-ribosylation of Ras homologs, vimentin, and the microfilament linked ezrin/radixin/moesin (ERM) proteins [7,25]. ExoT is known to inactivate Ras-mediated phagocytosis through ADP-ribosylation of the SH2 domain of the ubiquitous upstream regulatory factors CT10 regulator of kinase protein (CRK) and the CRK-like protein (CRKL) [24]. Although affecting the host cell cytoskeleton by various means, neither ExoS nor ExoT have been shown to ADP-ribosylate actin [7,26]. Therefore, it was unexpected that AexT from *A. salmonicida* was reported to modify actin [20], which is a common target protein for toxins of other clades [16,17,21,27].

Given the implications of these pathogens it is of continued interest to characterize their T3SS effectors. The recent determination of the crystal structures of *Pseudomonas* ExoS and ExoT ART domains [28] provided a new opportunity to address this aim. Here, we modelled the structure of the ART domain of AexT and predicted its target protein binding surface to be more similar to that of ExoT. This observation prompted us to purify full length AexT and ExoT proteins and to analyze their ADP-ribosyltransferase activities toward different target proteins. We present evidence supporting previous findings that both exotoxins catalyze arginine-linked ADP-ribosylation. We found that both AexT and ExoT modified actin, previously the only known target for AexT. Additionally, both AexT and ExoT modified the SH2 domain of CRKL, previously the only known target for ExoT. Thus, a common in vitro target specificity could be inferred from conserved surface features of the two exotoxins. Together, these results expand the known repertoire of virulence mechanisms in *P. aeruginosa* and *A. salmonicida* that lead to disruption of the cytoskeleton in infected host cells.

## 2. Results

### 2.1. Structural Model of AexT

Phylogenetic analyses indicated a close relationship between AexT and the more well-characterized 14-3-3 dependent ART toxins [21]. Alignment of the C-terminal sequences of AexT, ExoS and ExoT illustrates the near-perfect conservation of the 14-3-3 interaction sites (Figure 1a). These consist of (i) a recently discovered extensive hydrophobic surface patch with contributions of residues V414-LLSA(×8)RV-L429 (AexT sequence numbering); and (ii) the two “LDLA” boxes that interact with the phosphopeptide binding grooves in each monomer of the 14-3-3 dimer [28]. This sequence conservation confirms that AexT is a member of the 14-3-3 dependent ART toxins. The AexT GAP domain shares lower sequence homology with ExoS and ExoT than the latter share with one another. Although all three exotoxins share more than 80% homology within their ART domains, the AexT ART domain is more similar to ExoT than to ExoS (Figure 1b).

To explore the phylogenetic relationship within this clade of ART toxins for new insights into AexT functions, we constructed a homology model of the ART domain of AexT based on our recent crystal structure of the 14-3-3β:ExoT ART domain complex [28]. The quality of the model is good, with 2% of side chains in disallowed regions of the Ramachandran plot and an overall quality score of 0.68 determined by the QMEAN server [29] (Appendix A). The model aligns with a root mean square deviation of 1.4 Å over all Cα-atom pairs with a model produced by AlphaFold [30] (Appendix A). A heterotrimer model of AexT with the 14-3-3β homodimer is illustrated in Figure 1c. The signature motif of toxins in this clade, the conserved R-S-E triad [21] within the dinucleotide binding site, is in the expected position in our model (residues R306, S364, E403 of AexT; Figure 1d). Additionally, the surface of the AexT model shares the hydrophobic surface features in the 14-3-3 binding site with ExoT and ExoS (Figure 1e) [28].

Analysis of the AexT surface residues that are conserved with either ExoT or ExoS provided an important clue to shared functions among the three toxins: The overall number of conserved residues is similar in both pairwise comparisons; however, there is a prominent ridge of surface residues near the NAD+ binding site that is strictly conserved between AexT and ExoT, but divergent between AexT and ExoS (Figure 1f,g). This surface region coincides well with the surface that has been implicated in target protein recognition [19,31]. Furthermore, we noted that the canonical ExE motif, involved in catalysis in most ADP-ribosylating toxins, appears to be extended to 398ExxExE403 (AexT numbering) in AexT and ExoT exclusively. Fehr and co-workers found that the first and second, rather than second and third, glutamate of the motif were essential for catalysis in AexT (residues E401 and E403; Figure 1d) [20]. The conservation of these extended motifs among AexT and ExoT may be related to a shared target spectrum. In summary, structural modeling suggested a near homology of target recognition sites in AexT and ExoT, which was not anticipated previously. This prompted us to compare target protein ADP-ribosylation catalyzed by recombinant AexT and ExoT.

**Table 1 pathogens-11-01497-t001:** Multimeric states of exotoxin-chaperone complexes in solution.

		AexT-ART: 14-3-3β	FL-AexT: 14-3-3β	ExoT-ART: 14-3-3β	FL-ExoT: 14-3-3β	14-3-3β
Theoretical MW (kDa)	Exotoxin	27.503	52.658	26.689	51.067	
	14-3-3β *	28.534	29.356	28.534	29.356	29.356
	Complex	84.571 **	111.370 **	83.757 **	109.779 **	58.712 ***
Measured MW (kDa) ****	Mass Photometry	85 ± 19	106 ± 23	81 ± 18.8	107 ± 69	61 ± 12.8
	SEC-MALS	82.323	ND	82.352	ND	57.506

* With or without affinity tag, depending on protocol (see Materials and Methods); ** Assuming heterotrimer of two 14-3-3β molecules and one toxin molecule; *** Assuming homodimer; **** Means ± S.D. ND, not determined.

### 2.2. Recombinant Production of Full-Length Exotoxins

Exotoxin target protein modification has previously been investigated for full-length recombinant AexT [20] and both the recombinant ExoT ART domain in isolation and the full-length toxin secreted into mammalian cell cultures. [24,26,32] To achieve a comparative analysis, we determined that it was necessary to establish protocols for the recombinant production of the full-length and ART domain constructs of both AexT and ExoT. While the ART domains could be produced to satisfactory quantity and purity by purification from the soluble fractions of *E. coli* lysates, the full-length toxins were purified by refolding the proteins from the insoluble fractions (Appendix A). Purification of functional ExoT, as judged by its NAD+ glycohydrolase activity, was strongly aided by addition of purified 14-3-3β protein in excess (Appendix A) and therefore, 14-3-3β protein was added also during the refolding of AexT.

Recombinant ExoT and ExoS ART domains in complex with 14-3-3β were previously shown to form both heterotrimers and heterotetramers. [28] Here, we employed mass photometry to analyze the solution properties of the purified proteins. The results indicated that both ART domains and full-length exotoxins formed mainly heterotrimers with 14-3-3β (Figure 2a–d and Table 1). Heterotrimer formation was confirmed by analytical size exclusion chromatography in case of the ART domains (Figure 2f,g and Table 1). Full-length exotoxin complexes could not be assessed by this method, and exotoxin constructs in the absence of 14-3-3 protein could not be assessed by either of the two methods, due to extensive protein aggregation during the experiment. Using mass photometry at ~15 nM protein concentration, heterotrimers were apparently unstable, as indicated by the presence of the 57-kDa 14-3-3β homodimer peaks (Figure 2a–e). In contrast, in analytical size exclusion chromatography, the peak concentration was estimated to ~6 µM. Under these conditions, single peaks were observed indicating stable complexes. Together, these results suggest that *K*_D_ values describing the affinities of AexT and ExoT for 14-3-3β homodimers would lie in the mid-nanomolar range. For comparison, the affinity of the homologous ExoS-ART domain for 14-3-3β homodimers has been determined to a *K*_D_ of roughly 40 nM [28].

**Figure 2 pathogens-11-01497-f002:**
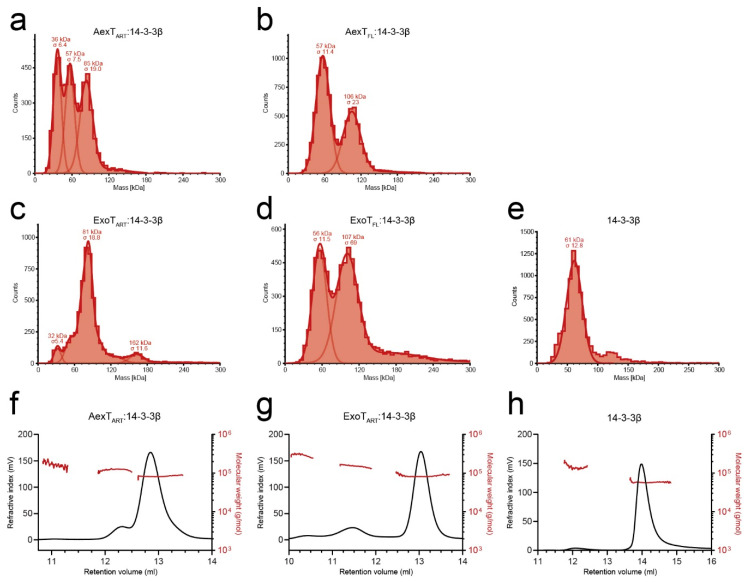
Solution properties of the recombinant exotoxins. (**a**–**e**) Size distributions of the 14-3-3β complexes of AexT-ART (**a**) and full-length (**b**) as well as ExoT-ART (**c**) and full-length proteins (**d**) measured using mass photometry. (**f**–**h**) Size distributions of the 14-3-3β complexes of AexT-ART (**f**) and ExoT-ART (**g**) assessed by analytical size exclusion chromatography. Size determinations of 14-3-3β homodimers by each method are shown for comparison (**e**,**h**). ART, ADP-ribosyltransferase domain; FL, full-length protein.

**Figure 3 pathogens-11-01497-f003:**
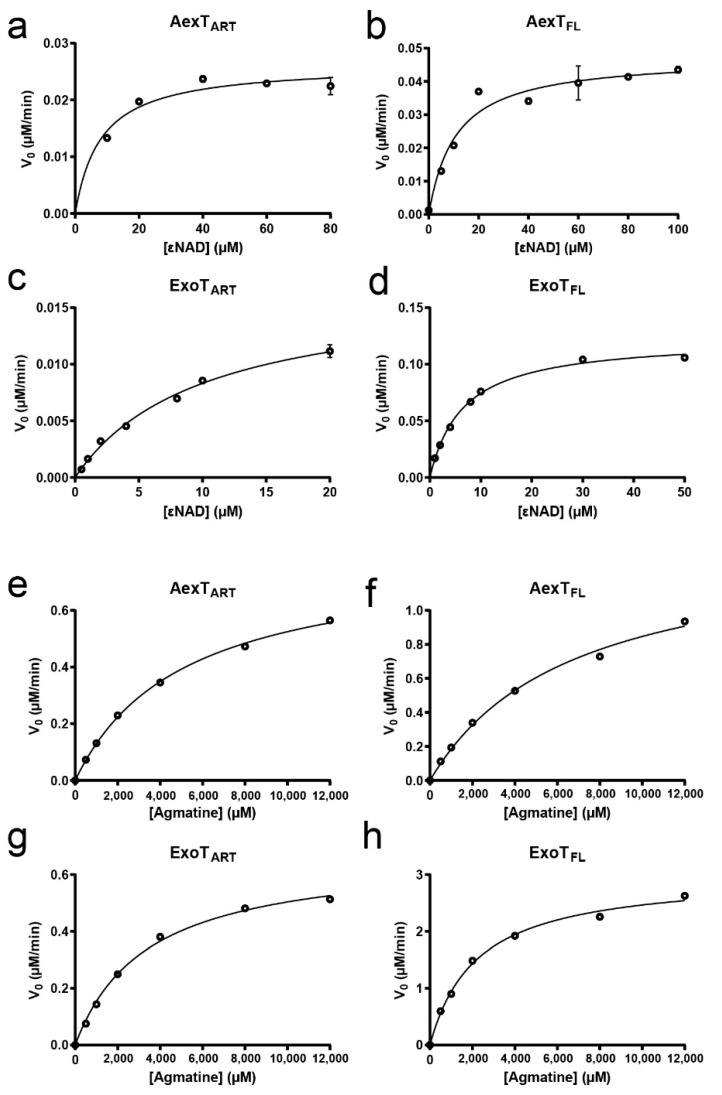
Steady-state kinetic parameters of the recombinant exotoxins. (**a**–**d**) Co-substrate dependent N^6^-etheno-NAD+-glycohydrolase activities of the indicated exotoxins, in presence of 14-3-3β. (**e**–**h**) Agmatine dependent N^6^-etheno-NAD+-glycohydrolase activities of the indicated exotoxins, in presence of 14-3-3β. The enzymatic parameters derived from these experiments are given in Table 2. Representative experiments are shown (n = 3).

### 2.3. Enzymatic Properties of AexT and ExoT

To establish basic enzymatic properties of our AexT and ExoT constructs, we employed an assay of NAD+ glycohydrolase activity in which the dinucleotide analog εNAD+ was used as co-substrate, resulting in a fluorescence increase upon its hydrolysis to εADP-ribose [33]. The result indicated that, in the absence of target protein (i.e., when εNAD+ was either hydrolyzed or processed in an auto-ADP-ribosylation reaction), both AexT and ExoT processed εNAD+ with a *K_M_* value in the low micromolar range (Figure 3 and Table 2). This was independent of whether the full-length proteins or the isolated catalytic domains were analyzed and indicated a dinucleotide affinity in the range of published values for other ADP-ribosylating bacterial toxins using the same assay (Table 2) [34]. Next, we analyzed AexT and ExoT activities in presence of agmatine, an analog for arginine [35]. Addition of agmatine led to a 20- to 30-fold increase in *k_cat_*. These results confirm previous analyses [20,36] showing that both AexT and ExoT can modify arginine residues (Figure 3 and Table 2).

**Figure 4 pathogens-11-01497-f004:**
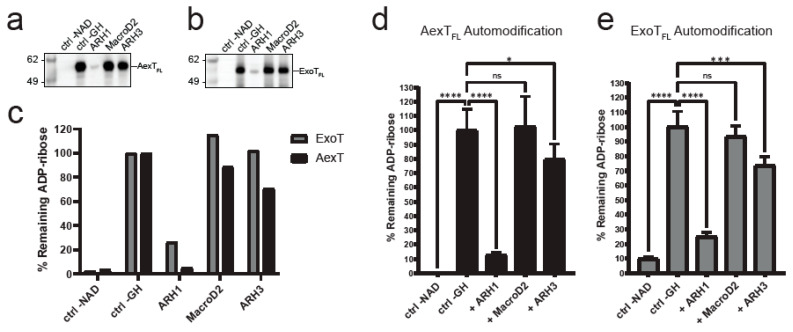
Exotoxin ADP-ribosylation and its removal by ADP-ribosylhydrolases. (**a**,**b**) Exotoxins were incubated with biotin-NAD+ and subsequently with the ADP-ribosylhydrolases indicated. Membranes were probed with streptavidin-HRP. (**a**) AexT; (**b**) ExoT. Uncropped images and loading controls are shown in Appendix A. (**c**) Quantification of exotoxin ADP-ribosylation from the membranes shown above. (**d**,**e**) Plate reader-based overlay assay using MacroGreen (see Materials and Methods). Shown are quantification of GFP signal indicating ADP-ribosylation after incubation with the ADP-ribosylhydrolases indicated. Significance levels: *p* ≥ 0.05 (ns), *p* < 0.05 (*), *p* < 0.001 (***), *p* < 0.0001 (****); n = 8. ART, ADP-ribosyltransferase domain; FL, full-length protein.

We wanted to further validate these findings by analyzing whether arginines were the only target residues for AexT and ExoT. To do this, we employed the residue specific ADP-ribosylhydrolases ARH1, ARH3, and MacroD2. We set up reactions that enable NAD+glycohydrolase and automodification activities of full-length exotoxins. We included 10% N^6^-biotin-NAD+ in the reaction buffer to analyze the results by Western blotting and detection using streptavidin-HRP. We analyzed exotoxins treated either with the arginine-ADP-ribosyl specific ADP-ribosylhydrolase ARH1 [37] or with the carboxyl-ADP-ribosyl specific MacroD2, [38] or with the serine-ADP-ribosyl specific ARH3 [39]. This revealed that AexT was capable of efficiently modifying itself, which has not been documented before (Figure 4a). Furthermore, ARH1 efficiently removed ADP-ribosyl from both AexT and ExoT, suggesting that automodification in both exotoxins occurred primarily at arginine residues (Figure 4). Neither ARH3 nor MacroD2 activities resulted in major reduction of ADP-ribosylation levels. However, ARH3 treatment gave a weak but significant reduction in ADP-ribosyl levels on both exotoxins. These results were confirmed by the MacroGreen assay (Figure 4d,e) that measures binding of an engineered Af1521 macrodomain [40], fused to GFP [41], to ADP-ribosylated protein immobilized on a microtiter plate. We conclude that both AexT and ExoT ADP-ribosylate themselves primarily on arginine residues.

**Figure 5 pathogens-11-01497-f005:**
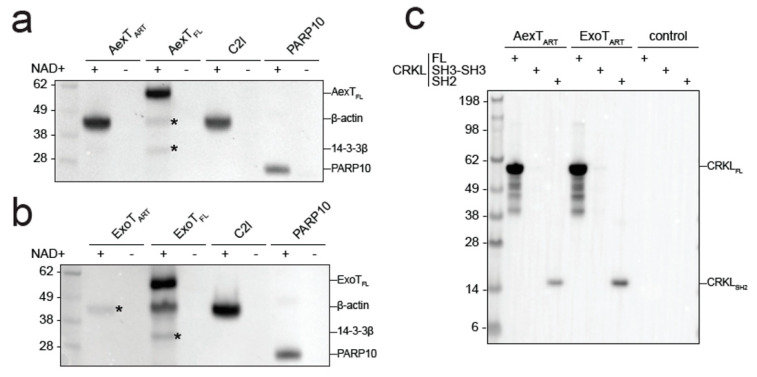
ADP-ribosylation of canonical AexT and ExoT targets by both exotoxins. (**a**,**b**) Exotoxin constructs were incubated with biotin-NAD+ and purified β-actin. Membranes were probed with streptavidin-HRP. C2I toxin, a known actin specific ADP-ribosyltransferase, and PARP10 ART domain, a strongly auto-ADP-ribosylating enzyme, were included for comparison. (**c**) ADP-ribosylation of CRKL constructs by exotoxin ART domains. Reactions were processed as in the previous panels. Uncropped images and loading controls are shown in Appendix A. ART, ADP-ribosyltransferase domain; FL, full-length protein.

Since AexT and ExoT have been reported to modify different target proteins, we sought to confirm those observations by analyzing ADP-ribosylation of purified AexT and ExoT target proteins by Western blotting. Both AexT (Figure 5a) and ExoT (Figure 5b) ADP-ribosylated actin, albeit with lower efficiency than the *Clostridium* C2I subunit. We observed that the isolated ART domain of AexT was more efficient at modifying actin than the full-length toxin, while the relationship was reversed for ExoT. We cannot exclude that low levels of actin modification by full-length AexT were caused by misfolding of the re-folded toxin; but this appears as an unlikely explanation in the light of the efficient automodification of AexT (Figure 4a and Figure 5a). Actin modification might still be significant in infected cells, where the ratio of actin over AexT is likely much higher than under our experimental conditions. The experiments shown in Figure 5a,b also reveal that AexT and ExoT automodification occurs only in the full-length exotoxins. This may explain the differences in NAD+ glycohydrolase rates observed between the ART domain and full-length constructs (Table 2): Whereas the ART domains catalyze NAD+ glycohydrolysis only, the full-length exotoxins catalyze both the glycohydrolysis reaction and their automodification. Interestingly, *Pseudomonas* ExoS automodification occurs in its GAP domain and suppresses the activity of that domain [42]. Our results imply that AexT and ExoT might employ the same mechanism of autoregulation.

AexT has been suggested to target actin exclusively [20] and be unable to modify the CRK proteins, which are known ExoT targets. However, we show here that both toxins were able to modify the SH2 domain of the CRK-like protein (CRKL) and the modification appeared to be more efficient when the SH2 domain was presented in the context of the full-length protein (Figure 5c). This experiment was carried out with the ART domain constructs in order to avoid overlap of automodified full-length toxins with the CRKL constructs, which have similar mobility in SDS-PAGE (see loading controls in Appendix A
Appendix A).

**Figure 6 pathogens-11-01497-f006:**
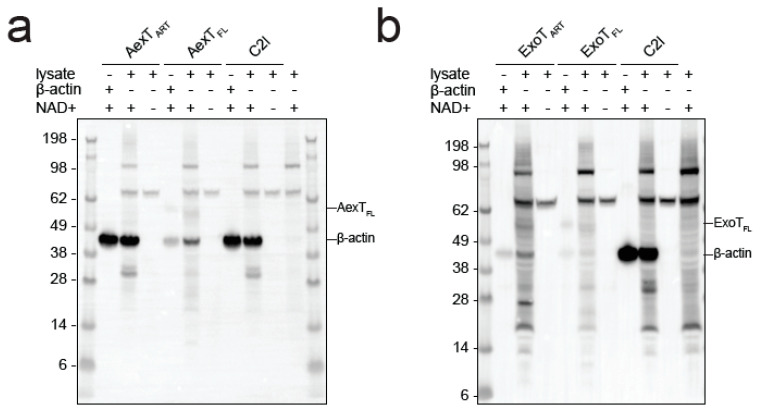
ADP-ribosylation of actin in THP-1 cell lysates. Exotoxin:14-3-3 complex or C2I and excess biotin-NAD+ were added to cell lysates. The reactions were processed by Western blotting followed by detection using streptavidin-HRP. (**a**) AexT constructs; (**b**) ExoT constructs. Uncropped images, alternative exposures and loading controls are shown in Appendix A. ART, ADP-ribosyltransferase domain; FL, full-length protein.

Modification of actin has not been reported for ExoT, to the best of our knowledge. Therefore, we explored this observation further. When incubated with cell lysates, ExoT as well as AexT and C2I modified a protein with a mobility in SDS-PAGE indicative of actin (Figure 6). All ADP-ribosylating toxins analyzed so far modify actin at residue R177 [43] and AexT is no exception [20]. ADP-ribosylation at R177 is known to block actin polymerization [44]. To test the effect of AexT and ExoT on the extent of actin polymerization, we conducted a sedimentation assay that is illustrated in Figure 7a. While actin ADP-ribosylation by C2I abolished actin polymerization, ADP-ribosylation by both AexT and ExoT led to partial inhibition of polymerization (Figure 7b). AexT and ExoT ART domain constructs were more efficient than the respective full-length proteins in preventing actin polymerization. This can likely be explained by the fact that actin ADP-ribosylation competed with automodification only in the case of the full-length exotoxins (Figure 5a,b). Nevertheless, these results show that AexT mediated ADP-ribosylation of actin compromises actin polymer formation. Furthermore, they suggest that ExoT can ADP-ribosylate actin on the same target residue as AexT and toxins from other clades.

## 3. Conclusions

We show that *Pseudomonas* ExoT and *Aeromonas* AexT are homologous toxins that share actin and CRK-proteins as targets *in vitro*. The existence of common targets for these two exotoxins was suggested by homology modeling (Figure 1) that placed conserved AexT residues in the experimentally determined target recognition site of ExoT [31]. Our data suggest a new mechanism for the ADP-ribosyltransferase activitiy of each toxin: ExoT might modify cellular actin to prevent polymerization; and AexT might affect the integrity of the microfilament system via CRK proteins and possibly contribute to induction of apoptosis. Therefore, it will be important to re-examine the ADP-ribosyltransferase activities of ExoT and AexT to test these findings in models of *Pseudomonas* and *Aeromonas* infection. If inhibitors of the exotoxin:14-3-3 interaction can be developed, they will be helpful tools to test exotoxin effects on host cell invasion and spreading of the pathogens within infected tissue. This would contribute to clarification whether these exotoxins are feasible targets toward development of therapeutic agents.

## 4. Materials and Methods

### 4.1. Materials

Fine chemicals and growth media were purchased from Merck Life Science AB, Solna, Sweden. 1,N^6^-biotin-NAD+ (biotin-NAD+) was obtained from BPS Bioscience, San Diego, Ca, USA, and 1,N^6^-etheno-AMP (εAMP) and -NAD+ (εNAD+) were obtained from Merck Life Science AB, Solna, Swedenand Jena Bioscience, Jena, Germany, respectively.

### 4.2. Structural Homology Model of AexT

The AexT, ExoT and ExoS sequences were downloaded from Uniprot (accession numbers Q93Q17, Q9I788, and G3XDA1, respectively) and aligned in Jalview [45] with the ClustalO algorithm. Given the sequence similarities within the ART domains, we used the structure of ExoT-ART:14-3-3β (PDB: 6GNN) [28] as a template and modelled the AexT-ART domain structure using SWISS-MODEL [46]. Model quality was assessed using Molprobity [47], which resulted in a score of 2.08, a clash score of 2.02, and 4.08% outliers in the Ramachandran plot. The model was superimposed with the ExoT-ART:14-3-3β structure using Chimera 1.16 [48].

### 4.3. Molecular Cloning

The cDNA fragment encoding the ART domain of AexT (AexT-ART; residues 252–475), codon-optimized for expression in *Escherichia coli* strains, was obtained from GeneArt (Thermo Fisher Scientific, Gothenburg, Sweden). For co-expression with 14-3-3β, the fragment was sub-cloned into the NcoI and EcoRI sites of a modified pET vector to obtain an N-terminal hexahistidine fusion. The ExoT ART domain (ExoT-ART; residues 235–457) sub-cloned in the corresponding vector was described earlier. [28] The hexahistidine tag on the ART domains enabled co-purification of un-tagged 14-3-3β in the toxin chaperone complex. The cDNAs encoding full-length AexT and ExoT were amplified by PCR from whole cell lysates of *Aeromonas salmonicida* strain A449 and *Pseudomonas aeruginosa* strain PAK, respectively. These cDNAs were inserted into pNIC28 (GenBank: EF198106) using ligase independent cloning. A cDNA encoding *Clostridium botulinum* C2I toxin subunit (residues 6–428), codon-optimized for expression in *E. coli* strains, was obtained from GeneArt (Thermo Fisher Scientific, Gothenburg, Sweden) and sub-cloned into the NcoI and BamHI sites of pET28 to obtain an N-terminal hexahistidine fusion. Human CRKL GST fusion plasmids (pGEX-2T) coding for either the SH2-SH3-SH3 domains, residues 7–303 or the SH3-SH3 domains, residues 115–303 were obtained from Addgene (#36400 and #36403). [49,50] Expression vector pNIC-H102 encoding the SH2 domain (residues 1–98) with an N-terminal decahistidine tag were obtained from Open Biosystems, Huntsville, Ga, USA (OHS4902). The expression plasmids for human 14-3-3β, ARH1, ARH3, MacroD2 and PARP10 have been described [28,41].

### 4.4. Protein Expression and Purification

Bovine cytosolic actin (β- and γ-isoform mixture) was purified from calf thymus and the isoforms were separated by chromatography on hydroxyapatite as described [51]. All other proteins were expressed in either *E. coli* BL21(DE3)T1R (Merck Life Science AB, Solna, Sweden) carrying the pRARE2 plasmid (Karolinska Institutet Protein Science Facility) or BL21(DE3) (New England Biolabs, C2530H). Expression cultures were set up in 2 L Schott flasks containing 1.5 L Terrific Broth supplemented with the appropriate antibiotics. Cells were grown at 37 °C in a LEX Bioreactor (Epiphyte3) until OD_600_ reached 2. The temperature was reduced to 18 °C and protein overexpression was induced with 0.5 mM IPTG for 16 h. Cells were pelleted by centrifugation at 4600× *g* at 4 °C and resuspended in lysis buffer (2.5 mL per gram of wet cell weight; 50 mM HEPES-NaOH pH 7.5, 500 mM NaCl, 10% glycerol, 1 mM TCEP supplemented with one tablet protease inhibitor cocktail (cat. no. S8830; Merck Life Science AB, Solna, Sweden,) and 5 µL benzonase (cat. no. E1014; Merck Life Science AB, Solna, Sweden). The suspension was sonicated on ice (Sonics VibraCell; 100 W with a 0.5-inch tip) for 4 min in 10 s intervals and subsequently centrifuged at 22,000× *g* for 25 min at 4 °C. The supernatant was clarified using a 0.45 µm filter prior to chromatography.

Immobilized metal affinity chromatography (IMAC) was carried out using 5 mL HiTrap TALON columns (Cytiva, Uppsala, Sweden) on an ÄKTA Pure system. The bound proteins were eluted with 50 mM HEPES-NaOH pH 7.5, 500 mM NaCl, 300 mM imidazole, 10% glycerol, and 1 mM TCEP over a 10-column volume gradient. Peak fractions were analyzed by SDS-PAGE (NuPage 4–12% Bis-Tris gels; Invitrogen, Thermo Fisher Scientific, Gothenburg, Sweden) and Coomassie staining [52] before subsequent purification steps.

AexT-ART:14-3-3β complex was further purified by cation exchange chromatography. Pooled IMAC fractions were desalted and injected in a 5 mL HiTrap Heparin HP column (Cytiva, Uppsala, Sweden) equilibrated in 20 mM MES pH 6.0, 1 mM MgCl_2_, 1 mM TCEP. The bound proteins were eluted with a 10-column volume gradient of the same buffer but containing 1 M NaCl.

For all proteins (except for full-length exotoxins; see 4.5), size exclusion chromatography (SEC) was carried out using a HiLoad 16/600 Superdex 75 pg column (Cytiva, Uppsala, Sweden) equilibrated in 50 mM HEPES-NaOH pH 7.5, 300 mM NaCl, 10% glycerol, 0.5 mM TCEP. Individually expressed 14-3-3β was further purified by hydrophobic interaction chromatography as described [28]. Protein purity was verified by SDS-PAGE and pure fractions were pooled and concentrated with Vivaspin centrifugal concentrators (cat. no. Z614025 or similar Merck Life Science AB, Solna, Sweden). Samples were frozen in aliquots and stored at −80 °C.

### 4.5. Protein Recovery from Inclusion Bodies

For full-length AexT and ExoT, protein overexpression and bacterial cell lysis were performed as described above. The supernatant insoluble fraction containing the inclusion bodies was resuspended in wash buffer A (100 mM Tris-HCl pH 7.5, 500 mM NaCl, 2 M Urea, 10 mM EDTA, 5 mM TCEP, 2% Triton X-100; 5 mL per gram of wet cell weight) using a glass-glass tissue homogenizer. The suspension was centrifuged at 15,000× *g* for 15 min at 4 °C to pellet inclusion bodies. This washing step was performed three times. Then, the pellet was resuspended in wash buffer B (100 mM Tris-HCl pH 7.5, 10 mM EDTA; 5 mL per gram of wet cell weight). The suspension was centrifuged at 15,000× *g* for 10 min at 4 °C, and the supernatant was discarded. The pellet was resuspended in solubilization buffer [6.6 M guanidine hydrochloride (GdnHCl), 50 mM Tris-HCl pH 8; 1 mL per 20–40 mg inclusion bodies], using a tissue homogenizer. The suspension was incubated for 2–3 h at room temperature under constant stirring. Cell debris and non-solubilized material were removed by centrifugation at 50,000× *g* for 20 min. For ExoT, the supernatant was incubated with Ni-NTA agarose (Qiagen, Kista, Sweden) at room temperature overnight under gentle shaking. The beads were collected by low-speed centrifugation and washed twice with 6 M GdnHCl, 50 mM Tris-HCl pH 8 and once with 6 M GdnHCl, 50 mM Tris-HCl pH 8.0, 12 mM imidazole. ExoT was eluted in 6 M GdnHCl, 50 mM Tris-HCl pH 8.0, 400 mM imidazole and the Ni-NTA bead eluate was diluted to a final concentration of 0.1 mg/mL with 6 M GdnHCl, 50 mM Tris-HCl pH 8.0. For the AexT-containing supernatant, the protein concentration was kept at 2 mg/mL. Both proteins were dialyzed at room temperature under gentle stirring, using high retention seamless cellulose tubings (14 kDa cut-off; cat. no. D0405; Merck Life Science AB, Solna, Sweden). Each dialysis step was performed in approximately 80 mL buffer per ml of protein solution for at least 2h, using the following buffers with decreasing GdnHCl concentration: Buffer A (4 M GdnHCl, 50 mM Tris-HCl pH 7.5), buffer B (2 M GdnHCl, 50 mM Tris-HCl pH 7.5), buffer C (0.5 M GdnHCl, 50 mM Tris-HCl pH 7.5), buffer D (50 mM Tris-HCl pH 7.5, 300 mM NaCl, 10% glycerol, 0.5 mM TCEP). Before starting the final dialysis step, purified 14-3-3β was added in approximately equimolar concentrations to the exotoxins to assist refolding. The refolded proteins were separated from aggregates by centrifugation at 4000× *g* for 5 min and SEC was performed as above but sing a HiLoad 16/600 Superdex 200 pg column (Cytiva). Protein purity was verified by SDS-PAGE and pure fractions were pooled and concentrated with Vivaspin 20 centrifugal concentrators (30 kDa MWCO, cat. no. GE28–9323–61, Merck Life Science AB, Solna, Sweden). The proteins were frozen in aliquots and the degree of refolding was evaluated by measuring enzymatic activity.

### 4.6. Analysis of Protein Solution Properties

The oligomeric states of the exotoxin:14-3-3β complexes were analyzed using mass photometry on a REFEYN TwoMP mass photometer. Microscopy slides (24 × 50 mm) were rinsed consecutively with milli-Q water, isopropanol, and milli-Q water and dried under a clean nitrogen stream. A row of 2 × 3 silicone gasket wells (Grace Biolabs, Merck Life Science AB, Solna, Sweden) was placed onto a slide. Protein batches were thawed and diluted to approximately 100 nM in buffer containing 20 mM HEPES-NaOH pH 7.5, 50 mM NaCl, 4 mM MgCl_2_, 0.5 mM TCEP. The focus was adjusted to a well containing 15 µL buffer before 3 µL of protein solution was added, yielding a final protein concentration of approximately 16 nM. The landing events within the recorded frame (900 × 354 pixels) were measured for 60 s at a frame rate of 496 Hz and at least 4900 events were recorded per acquisition. Data were analyzed using DiscoverMP software (REFEYN, Oxford, UK). For calibration, a native PAGE marker (Invitrogen, Thermo Fisher Scientific, Gothenburg, Sweden) was used. Oligomeric states were also characterized by analytical size exclusion chromatography (OMNISEC; Malvern Panalytical, Uppsala, Sweden) equipped with a Superdex 200 Increase 10/300 GL column (Cytiva, Uppsala, Sweden) equilibrated in 50 mM HEPES-NaOH pH 7.5, 300 mM NaCl and 1 mM TCEP. The detectors were calibrated with bovine serum albumin. Protein aliquots were filtered with 0.2 μm centrifugal filters and kept at 4 °C until injected into the column. Each sample was injected in duplicates of 120 µL, corresponding to 50–60 µg of protein per injection. Proteins were eluted at 0.5 mL/min. Data were analyzed using OMNISEC v11.32 software (Malvern Panalytical). One of two reproducible runs was chosen for analysis. The refractive index and measured molecular weight parameters were re-plotted in Prism (GraphPad Software LLC, DanDiego, CA, USA).

### 4.7. Enzymatic Analyses

In general, exotoxin assays contained 200 nM exotoxins and 2–2.5-fold molar excess of 14.3.3β in 20 mM HEPES-NaOH pH 7.5, 50 mM NaCl, 4 mM MgCl_2_, 0.5 mM TCEP. The source of 14-3-3β depended on the production protocol (see Section 4.3,Section 4.4 and Section 4.5). In reactions containing agmatine (Merck Life Science AB, Solna, Sweden), εNAD+ was kept at 25 µM. For negative control reactions, εNAD+ or toxin was omitted. Measurements were carried out in black flat-bottom half-area non-binding 96-well plates (Corning CLS3993; Merck Life Science AB, Solna, Sweden) in final volumes of 50 µL. Enzymatic reactions, at room temperature (ca. 22 °C), were started by the addition of εNAD+, and fluorescence was followed over time in a CLARIOstar multimode plate reader (BMG Labtech, Ortenberg, Germany) using λ_ex_ = 302/20 nm (filter) and λ_em_ = 410/20 nm (monochromator). All kinetic parameters were determined based on fluorescence increase in the linear time range. Fluorescence intensities were related to concentrations of the fluorescent product using dilution series of εAMP. Kinetic parameters were calculated using Prism (GraphPad Software LLC, DanDiego, Ca, USA).

For detection of ADP-ribosylation by Western blotting, enzymatic reactions were performed with 100 µM NAD+ (Figure 5a,b and Figure 6) or 50 µM NAD+ (Figure 5c) containing 10% biotin-NAD+. Enzyme concentrations were 1 µM 14-3-3 dependent exotoxins with 2–2.5-fold molar excess of 14-3-3β, 20 nM *Clostridium botulinum* C2I toxin subunit, and 2 µM PARP10 catalytic domain. For the experiments depicted in Figure 5c and Figure 6, the concentration of exotoxins was reduced to 0.5 µM. Target protein concentrations were 10 µM purified protein or 0.5 mg/mL total protein for reactions containing cell lysate. Reactions (20–40 µL in 20 mM HEPES-NaOH 7.5, 50 mM NaCl, 4 mM MgCl_2_, 0.5 mM TCEP) were started by the addition of either NAD+ or toxin. Reactions were incubated for 1 h at room temperature and stopped by addition of Laemmli buffer and heating at 95 °C for 4 min. Automodification reactions prior to ADP-ribosylhydrolase treatment were conducted essentially as above, with 1 µM 14-3-3 dependent exotoxins, 2–2.5-fold molar excess of 14-3-3 β and 5 µM NAD+. After 1 h of incubation, the reactions were stopped by addition of 100 µM inhibitor STO1101 [53]. After 20 min of incubation at room temperature, (ADP) ribosylhydrolases ARH1, MACROD2 and ARH3 were added at 10 µM, followed by incubation for 2 h. Reactions were stopped with Laemmli buffer and heating at 95 °C for 4 min. Proteins were separated on 4–12% Bis-Tris polyacrylamide gels (Invitrogen, Thermo Fisher Scientific, Gothenburg, Sweden) and transferred onto a PVDF membrane in transfer buffer (3% (m/v) Trizma, 14.4% (m/v) glycine and 10% (v/v) methanol) for 1h at 160 mA. Ponceau-S staining of the membrane served as a control for loaded protein and transfer efficiency. Membranes were de-stained, blocked for 1 h in 1% BSA in TBST, and incubated for 1 h in 0.5 µg/mL HRP-linked Streptavidin (cat. no. 21126; Thermo Fisher Scientific, Gothenburg, Sweden) in 1% BSA in TBST. Proteins modified with biotin-NAD+ as co-substrate were visualized with SuperSignal™ West Pico PLUS Chemiluminescent Substrate (Thermo Fisher Scientific, Gothenburg, Sweden). Images were obtained using a ChemiDoc imaging system (Bio-Rad, Solna, Sweden) and densitometric analysis was performed in Image Lab software (Bio-Rad, Solna, Sweden).

Target residue specificity of the exotoxins was also determined using a macrodomain-GFP fusion overlay assay [41]. Auto-ADP-ribosylation reactions of 250 nM exotoxin and 2-2.5-fold molar excess of 14-3-3β were carried out in reaction buffer (25 mM HEPES-NaOH pH 7.5, 100 mM NaCl, 0.2 mM TCEP, 4 mM MgCl_2_) in the presence of 1 mM NAD+ at room temperature for 30 min. The reactions were transferred into a 96-well high-binding plate (Nunc MaxiSorp™; Thermo Fisher Scientific, Gothenburg, Sweden). Proteins were allowed to bind to the plate for 30 min. To remove NAD+, the wells were rinsed three times with reaction buffer and then blocked for 5 min with 1% BSA in reaction buffer. The wells were rinsed twice before ARH1, MacroD2, or ARH3 were added at 1.5 μM in reaction buffer. After 1h of incubation, the wells were rinsed twice with reaction buffer and then washed with 1 mM ADP-ribose in reaction buffer for 10 min. The wells were rinsed three times with TBST buffer, blocked again with 1% BSA in TBST buffer and rinsed twice. 1 μM of eAF1521-GFP [41] was added for detection of remaining ADP-ribose and three rinsing steps in TBST buffer were performed to remove unbound eAF1521-GFP. 150 µL TBST buffer was added to the wells for measurement. GFP fluorescence was measured on a CLARIOstar multimode reader (BMG Labtech, Ortenberg, Germany) using a filter set of λ_ex_ = 470/15 nm and λ_em_ = 515/20 nm. The data were normalized to the background signal (blank wells) and the percentage of remaining ADP-ribose was determined by comparison to auto-modification control reactions with no ADP-ribosylhydrolases added. The normalized data were plotted in Prism (GraphPad Software) and the significance was determined with a Welch ANOVA test, in which the mean of each column was compared to the mean of the auto-modification reaction.

### 4.8. Preparation of Cell Lysates

THP-1 human monocytic leukemia cells were grown in RPMI medium supplemented with 10% fetal bovine serum and 1% penicillin/streptomycin. 300x10^6^ cells were harvested and washed in 2 mL ice-cold D-PBS and subsequently lysed in 800 μL hypotonic buffer (50 mM Tris-HCl pH 7.5) supplemented with protease and phosphatase inhibitor cocktails (Complete, 11504400 and PhosSTOP EASYpack, 04906845001, respectively; Roche, Basel, Switzerland). The cells were resuspended by vortexing and pipetting thoroughly during 20 min incubation on ice. Protein concentration was determined using Bradford reagent (Bio-Rad). Before freezing, 80µL of 200 mM HEPES-NaOH pH 7.5, 500 mM NaCl, 40 mM MgCl_2_, 5 mM TCEP and 0.1% benzonase were added.

### 4.9. Actin Polymerization Assay

Bovine cytosolic β-actin in G-actin buffer (20 mM HEPES-NaOH pH 7.5, 0.2 mM CaCl_2_, 0.2 mM ATP, 0.5 mM TCEP) was thawed and diluted 1:1 in fresh G-actin buffer. The sample was centrifuged for 15 min at 21,000× *g*, 4 °C, and the supernatant was transferred to a new tube. 200 μM NAD+, 10 μM β-actin, and either 800 nM exotoxin:14-3-3β complex or 20 nM Clostridium botulinum C2I toxin subunit in G-actin buffer were added in 100 μL reactions and the samples were incubated for 30 min at room temperature. Actin polymerization was triggered by the addition of 100 mM KCl and 2 mM MgCl_2_. The samples were transferred into airfuge tubes (cat. no. 342630; Beckman Coulter, Bromma, Sweden). After 45 min at room temperature, actin polymers were sedimented by ultracentrifugation in a Beckman Coulter Airfuge (A-100/18 rotor) for 15 min at 30 psi (corresponding to roughly 149,000× *g*). The supernatant was separated from the pellet and the pellet was resuspended in 100 μL G-actin buffer. Equal volumes of supernatant and pellet fractions were loaded on a 4–12% NuPage Bis-Tris gel (Invitrogen, Thermo Fisher Scientific, Gothenburg, Sweden) and subjected to gel electrophoresis. The gel was stained with Coomassie.

## Figures and Tables

**Figure 1 pathogens-11-01497-f001:**
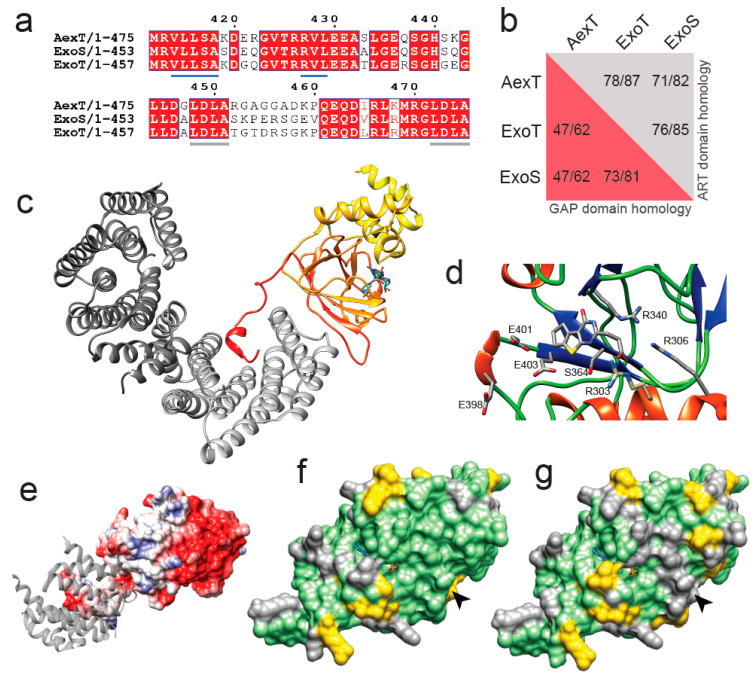
Homology model of the ART domain of *Aeromonas salmonicida* AexT. (**a**) Alignment of the C-terminal sequences of AexT and *Pseudomonas aeruginosa* ExoS and ExoT. The hydrophobic interaction site with 14-3-3 is underlined in blue and the LDLA-boxes are underlined in grey. (**b**) Extent of sequence homology (expressed as identity/similarity) between the GAP domains (red background) and ART domains (grey background) of the three exotoxins. (**c**) Homology model of the ART domain of AexT (color gradient from yellow (N-terminus) to red (C-terminus)), based on the ExoT structure (PDB: 6GNN). A carbaNAD+ molecule (sticks) was modeled based on the ExoS structure (PDB: 6GNK) to indicate the location of the active site. The 14-3-3β homodimer is shown in grey (different shades for each monomer). (**d**) Close-up of the active site containing ExoT inhibitor STO1101, with key side chains shown as sticks. (**e**) Electrostatic surface model of the AexT ART domain, showing the hydrophobic 14-3-3 binding site in light grey and a 14-3-3 monomer as grey ribbon. (**f**,**g**) Surface representation of the AexT ART domain colored for sequence conservation with either ExoT (**f**) or ExoS (**g**). Arrowheads indicate the location of the target binding sites (see text). Green, identity; yellow, similarity; grey, divergence.

**Figure 7 pathogens-11-01497-f007:**
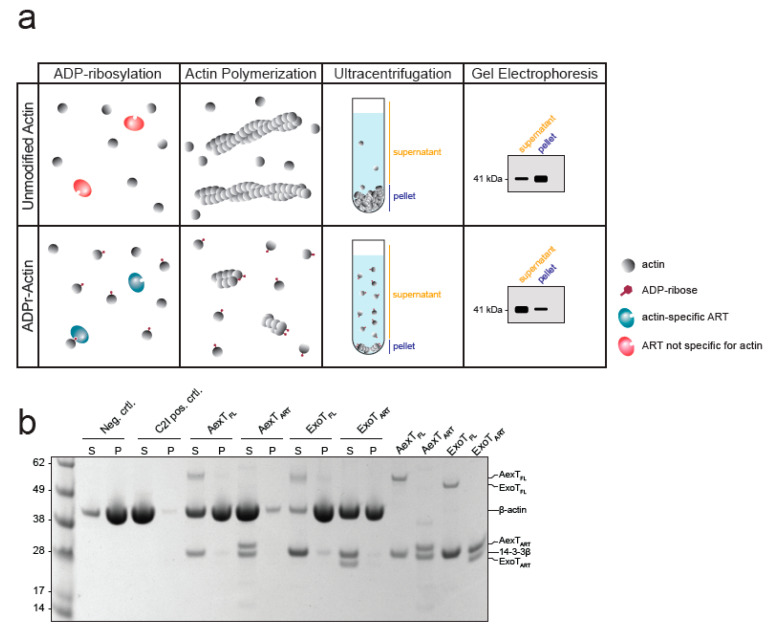
Exotoxin mediated modification compromises actin polymerization. (**a**) Schematic representation of the F-actin sedimentation assay employed. Exotoxin:14-3-3 complex and NAD+ were added to monomeric actin. After incubation to allow actin ADP-ribosylation, actin polymerization was induced and allowed to proceed to steady-state before actin polymers were sedimented. (**b**) The Coomassie stained SDS-PAGE gel shows equal amounts of pellet and supernatant fractions containing actin polymers and monomers, respectively. ART, ADP-ribosyltransferase domain; FL, full-length protein.

**Table 2 pathogens-11-01497-t002:** Steady-state kinetic parameters of the recombinant exotoxins.

Type of Reaction		NAD+Glycohydrolysis/Automodification	+Agmatine Modification
Exotoxin Construct ^1^	*K*_M_ (NAD+) ^2^	*k*_cat_ ^3^	*k*_cat_ ^3^
AexT-ART	8.1 ± 1.5	0.13	3.96
AexT-FL	10.9 ± 2.0	0.24	7.05
ExoT-ART	10.1 ± 1.0	0.09	3.37
ExoT-FL	6.6 ± 0.2	0.62	15.0

^1^ ART, ADP-ribosyltransferase domain; FL, full-length protein; both as 14-3-3β complexes. ^2^ in µM. ^3^ in min^−1^. Original data shown in Figure 3.

## Data Availability

Data is contained within the article or Appendix A.

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
