# Peer review of "14-3-3 Activated Bacterial Exotoxins AexT and ExoT Share Actin and the SH2 Domains of CRK Proteins as Targets for ADP-Ribosylation"

_pathogens, 2022, doi:10.3390/pathogens11121497_

Round 1
Reviewer 1 Report
The authors performed a structural study on ExoT and AexT exotoxins. They combined mathematical modeling and bioinformatics with protein purification and in vitro tests to prove that ExoT and AexT are homologous toxins that target actin and CRKL. The study is clearly written with convincing experimental work.
While the authors entirely focus on the descriptive and structural work, it would be beneficial to include functional analyses of the toxins. This would broaden the topic to a more general audience. It is notably not tested if deletion mutants ΔexoT and ΔaexT are less or avirulent in a cell culture model. This is a key information to understand to importance of the toxins in vivo and should be discussed in the introduction (as initiated at line 36) and conclusion parts. Authors should also include a paragraph concerning the genomic situation of the corresponding genes. Are these toxins on the chromosome or on a plasmid? Were they horizontally acquired? Are the toxins under any type of regulation and/or part of a specific regulon or operon?
Lines 53-58. The introduction of the two toxins ExoT and AexT is confusing and hard to read. Please rephrase this paragraph in a simpler way and describe each toxin one by one. The role of CRK and CRKL and the consequence of their modification is also lacking and would help to understand the function of the toxins. At line 53, what does co-express mean? Are exoS and exoT in an operon? What does it mean for their regulation and function?
Author Response
Response to Reviewer 1
We thank reviewer 1 who addressed our description of the subjects of our study and we agree that there was a need to improve their background description. We have re-written the introduction section taking care to include the aspects highlighted by the reviewer. We believe that the introduction is now easier to follow and will provide enough background information in order to understand the motivation for the experiments in the results section.
Point 1:
While the authors entirely focus on the descriptive and structural work, it would be beneficial to include functional analyses of the toxins. This would broaden the topic to a more general audience. It is notably not tested if deletion mutants ΔexoT and ΔaexT are less or avirulent in a cell culture model. This is a key information to understand to importance of the toxins in vivo and should be discussed in the introduction (as initiated at line 36) and conclusion parts.
We now added a better description of the importance of AexT, ExoT, and ExoS for the respective pathogens in infection and explained the observed associated morphological chances of cells infected with the exotoxins.
Point 2:
Authors should also include a paragraph concerning the genomic situation of the corresponding genes. Are these toxins on the chromosome or on a plasmid? Were they horizontally acquired? Are the toxins under any type of regulation and/or part of a specific regulon or operon?
Whereas not much is known about AexT in that aspect, Pseudomonas gene expression is much better understood. We added the relevant background information on the expression control of ExoS and ExoT as we considered that useful for our introduction. We describe that exoS and exoT, as part of the T3SS operon in Pseudomonas, are collectively controlled by the transcription factor ExsA.
Point 3:
Lines 53-58. The introduction of the two toxins ExoT and AexT is confusing and hard to read. Please rephrase this paragraph in a simpler way and describe each toxin one by one. The role of CRK and CRKL and the consequence of their modification is also lacking and would help to understand the function of the toxins. At line 53, what does co-express mean? Are exoS and exoT in an operon? What does it mean for their regulation and function?
This point is highly appreciated as after reflection on our introduction, we also found that non-experts will find it confusing and difficult to follow. We re-structured this section, including the one-by-one description of the known biochemical and structural properties as well as the target substrate specificity of the individual exotoxins. We also included an explanation of how the ADP-ribosylation of CRK proteins by ExoT is thought to affect the integrity of the infected cell. Finally, we clarified what co-expression of ExoT means (see point 2 above).
Reviewer 2 Report
The contribution by Ebenwaldner et al. reports a biochemical analysis of the activity of a bacterial exotoxin AexT from Aeromonas salmonicida, whose activity strongly resembles that of ExoT from Pseudomonas aeruginosa. Both proteins possess ADP-ribosylation activity that is dependent on binding to 14-3-3 proteins. Homology-based modeling of AexT to ExoT in the ExoT:14-3-3 protein complex reveals a high degree of similarity between the two bacterial exotoxins. The authors further examine the AexT:14-3-3 and ExoT:14-3-3 complexes by mass photometry and size exclusion chromatography and determine the steady-state parameters for the auto-ADP-ribosylation and ADP-ribosylation of agmatine. This arginine analog indicates that arginine residues are the likely targets for ADP-ribosylation by AexT and ExoT proteins, which was later confirmed by residue-specific removal of the modification by ARH1. In contrast to previous publications, the authors show that AexT and ExoT are also able to modify beta-actin and CRKL.
Overall, the experiments appear to be sound and support the authors' conclusions. However, some controls are missing, and certain parts of the text are unclear. For example, it is unclear how much new information this work actually contains, as the introduction is short and unfocused and the discussion is missing.
Major comments:
1. The introduction does not present sufficient information on the subject and is unfocused. It should be clear from the Introduction how much is actually known about A. salmonicida AexT and P. aeruginosa ExoT and what new information this work brings.
2. The information made clear by homology modeling of AexT appears to have no importance for the rest of the papers – there appears to be no follow up from this.
3. The analysis of the stoichiometry of AexT:14-3-3 and ExoT:14-3-3 protein complexes: controls are missing. Histograms and absorbance curves are missing for the individual proteins. In Table 1 masses of individual proteins are missing.
4. Neither the Results nor the Methods section are clear on how 14-3-3 proteins were added to the enzymatic mixtures (data in Fig. 3 and Table2). Since full length proteins were isolated from inclusion bodies and 14-3-3 proteins were used in refolding, were the latter removed after refolding. If not, how was the concentration of 14-3-3 proteins controlled in individual enzymatic mixtures (NB: separate domains were expressed as soluble proteins, so they are free of 14-3-3).
5. It is unclear what are the conclusions of the kinetic analyses (Fig 3, Table 2). What is “4-in mM” in the Table legend?
6. Some background o the GFP-based assay should be provided (lines 184-187).
7. Self standing exotoxin domains appear to inhibit actin polymerization more than full length proteins. Can the authors elaborate?
Minor comments:
Abstract: full names of the exotoxins and other proteins should be presented.
Introduction: Seafood, not “seefood”.
Line 53: references are duplicated (13, 14).
Lines 72 and 96: unnamed residues should be denoted either X or x, consistently.
Line 82: what makes the models “good”- a description is missing.
The legend of Figure 1: 14-3-3 protein coloring is missing.
Figure 3 title: “Steady state kinetic parameters of recombinant exotoxins” might be better suited.
Table 2: the title is not appropriate.
Line 153: highly diluted complex mixture leads to protein dissociation unless they are still above the Kd (has the Kd been determined in the literature?).
KM and kcat are constants and are typically italicized.
Line 164: this is likely Table 2, not 1.
Line 240: ribosylation [by] C2I.
Line 396: “KIinetic” – kinetic is misspelled.
Author Response
Response to Reviewer 2
We thank reviewer 2 and we appreciate the detailed and insightful comments, which have been very helpful in improving the quality of the manuscript.
Major comments
- The introduction does not present sufficient information on the subject and is unfocused. It should be clear from the Introduction how much is actually known about A. salmonicida AexT and P. aeruginosa ExoT and what new information this work brings.
This point is highly appreciated as we can appreciate that readers may find our introduction confusing and difficult to follow. We re-structured this section, starting with an introduction of the pathogens and their related diseases upon infection. We then introduce ExoS, ExoT and AexT and explain the roles these toxins play in the infection process of the respective pathogens. We then describe how ADP-ribosylation in infection is common among several pathogens before we explain the known biochemical and structural properties as well as the target substrate specificity of the three toxins in detail. Finally, we provide a summary of what we found during this study and give an outlook on why the characterization of ExoT and AexT is of continued interest. We believe that the new version is sufficiently detailed to both provide an introduction to the exotoxins’ functions in pathological mechanisms as well as specify the contributions of this study.
- The information made clear by homology modeling of AexT appears to have no importance for the rest of the papers – there appears to be no follow up from this.
The remainder of the paper probes the observations made by homology modelling so it is unfortunate that this relationship was not clear enough in our previous manuscript version. We have now re-written the last paragraph of the introduction section; the summary at the end of the modelling paragraph in the results section; and the conclusion section to clarify this issue.
- The analysis of the stoichiometry of AexT:14-3-3 and ExoT:14-3-3 protein complexes: controls are missing. Histograms and absorbance curves are missing for the individual proteins. In Table 1 masses of individual proteins are missing.
- 14-3-3b forms homodimers with high affinity and these homodimers coexist with the exotoxin:14-3-3 heterotrimers in the mass photometry experiments (conducted at low protein concentration where the 14-3-3 dimer is stable but the trimeric complex is unstable). Thus we agree with the reviewer that it is necessary to show the outcome of experiments in the absence of exotoxins as controls. We have now added these controls as panels e and h of Figure 2, and amended Table 1.
- Concerning the individual exotoxins; neither the ART domains nor the full-length proteins are stable enough under the experimental conditions and they aggregate in the absence of 14-3-3 proteins without giving meaningful information about their molecular sizes. We have added a statement explaining this to the end of section 2.2.
We agree that this section needed improvement and we hope that it now contains all the information essential for the reader.
- Neither the Results nor the Methods section are clear on how 14-3-3 proteins were added to the enzymatic mixtures (data in Fig. 3 and Table2). Since full length proteins were isolated from inclusion bodies and 14-3-3 proteins were used in refolding, were the latter removed after refolding. If not, how was the concentration of 14-3-3 proteins controlled in individual enzymatic mixtures (NB: separate domains were expressed as soluble proteins, so they are free of 14-3-3).
We have now extended the materials and methods section under points 4.3. (Molecular cloning), 4.4. (Protein expression and purification), and 4.5. (Protein recovery from inclusion bodies) to give a better explanation of the matter. In brief, ART domains were produced using a 14-3-3 co-expression vector and full-length toxins were purified after adding excess 14-3-3 to the refolding reactions. For enzymatic analyses, purified 14-3-3 was added to adjust the concentration to a 2.5 molar ratio to prevent loss of activity due to complex disassembly.
- It is unclear what are the conclusions of the kinetic analyses (Fig 3, Table 2). What is “4-in mM” in the Table legend?
Our most important conclusions from the kinetic analyses are two: (1) Both AexT and ExoT modify arginine residues. This conclusion should be clear from our description of the results. (2) The isolated ART domain activities (in the absence of target proteins) are restricted to NAD+ glycohydrolysis, whereas the full-length exotoxin kcat values reflect both NAD+ glycohydrolysis and automodification. We now discuss this point in the context of Figure 5, where we show the automodification of full-length exotoxins by Western blotting.
- Some background o the GFP-based assay should be provided (lines 184-187).
We have added a very brief expansion at this place in the manuscript (second paragraph of section 2.3) as well as added further details to the relevant section in Materials and Methods.
- Self standing exotoxin domains appear to inhibit actin polymerization more than full length proteins. Can the authors elaborate?
This is an important observation. We cannot be sure about the cause of this. However, we assume that the auto-modification activity of the full-length exotoxins reduces the trans-modification activity towards actin i.e. that actin is a less-preferred target compared to the GAP domain of the exotoxins and that this explains why the effect on actin polymerization is less prominent after modification with the full-length exotoxins. We have now added this suggestion in a discussion sentence in section 2.3.
Minor comments:
- Abstract: full names of the exotoxins and other proteins should be presented.
- Introduction: Seafood, not “seefood”.
- Line 53: references are duplicated (13, 14).
- Lines 72 and 96: unnamed residues should be denoted either X or x, consistently.
These errors have been corrected.
- Line 82: what makes the models “good”- a description is missing.
We have added a description of the model quality, and how it was assessed, in the description of the model in addition to presenting the results in the SI.
- The legend of Figure 1: 14-3-3 protein coloring is missing.
- Figure 3 title: “Steady state kinetic parameters of recombinant exotoxins” might be better suited.
- Table 2: the title is not appropriate.
These errors have been corrected.
- Line 153: highly diluted complex mixture leads to protein dissociation unless they are still above the Kd (has the Kd been determined in the literature?).
To our knowledge, this information is available for neither ExoT nor AexT. We have measured the affinities of ExoS ART domain constructs by fusing their N-termini to GFP and evaluating titration series with 14-3-3 by fluorescence anisotropy. This was done in the context of characterizing a novel 14-3-3 interaction site (Karlberg et al., 2018). In the present study, we do not believe that the in vitro Kd values of the exotoxins for 14-3-3 proteins are of relevance. However, for the purpose of clarifying the outcome of the measurements shown in Fig. 2, we agree that a brief discussion was helpful and we have added that to section 2.2.
- KM and kcat are constants and are typically italicized.
- Line 164: this is likely Table 2, not 1.
- Line 240: ribosylation [by] C2I.
- Line 396: “KIinetic” – kinetic is misspelled.
These errors have been corrected.
Round 2
Reviewer 2 Report
The revised version of the manuscript is appropriate.